# DNA quantification of basidiomycetous fungi during storage of logging residues

Isabella Børja, Gry Alfredsen, Tore Filbakk and Carl Gunnar Fossdal

Norwegian Forest and Landscape Institute, Ås, Norway

## ABSTRACT

The demand for bioenergy caused an increased use of logging residues, branches and treetops that were previously left on the ground after harvesting. Residues are stored outdoors in piles and it is unclear to what extent fungi transform this material. Our objective was to quantify the amount of wood degrading fungi during storage using quantitative real-time PCR (qPCR) to detect basidiomycetous DNA in logging residues, a novel approach in this field. We found that the qPCR method was accurate in quantifying the fungal DNA during storage. As the moisture content of the piled logging residues decreased during the storage period, the fungal DNA content also decreased. Scots pine residues contained more fungal DNA than residues from Norway spruce. Loose piles had generally more fungal DNA than bundled ones.

## INTRODUCTION

Logging residues, also known as forest slash, are branches and tops left on the forest site after logging. In Norway spruce (*Picea abies* (L.) Karst.) and Scots pine (*Pinus sylvestris* L.) residues correspond to about 55 and 20% of the stem volume, respectively (*Hakkila, 1991*). In the past, the logging residues were uneconomical to gather and were left in the forest. With the increased need to utilize all renewable sources of energy, this lignocellulosic biomass provides a new potential as a fuel, e.g., chips for waterborne heating. Because loose logging residues are bulky to handle, they are sometimes compressed into bundles and piled for easier handling and transportation (*Johansson et al., 2006*). To promote drying of the material, piles are left in the forest or at the roadside for variable periods of time until further processing to final biofuel. While storage is used as a method to reduce moisture and thus makes the material better suited as fuel, the material also undergoes a transformation as it decomposes. Fungi are known as the main degraders of woody materials (*Dix & Webster, 1995*). However, to our knowledge the fungal colonization of unprocessed logging residues, tops and twigs, has not been quantified.

The moisture content and temperature are essential factors for fungal growth. Fungal decomposers are divided into three functional groups according to their substrate utilization pattern: white-, brown- and soft-rot fungi (*Cooke & Rayner, 1984*). White-rot fungi are mainly basidiomycetes and the only organisms known to be able to effectively utilize the lignin, cellulose and hemicellulose in various proportions (*Cooke & Rayner, 1984*). Brown-rot fungi, which appear to be exclusively basidiomycetes, utilize cellulose

Corresponding author
Isabella Børja,
boi@skogoglandskap.no

and hemicellulose, leaving the modified lignin in place (*Cooke & Rayner, 1984*). Soft rot decay by ascomycetes and mitotic fungi primarily occurs under conditions where the growth of the generally more active and competitive basidiomycetes is retarded (e.g., high moisture, low aeration). The decay caused by soft rot fungi is generally slower than decay caused by basidiomycetes. Basidiomycetes are likely the fungal group most responsible for the logging residue degradation (*Tuomela et al., 2000*).

Traditionally, to detect fungal biomass in soil and plant materials, the classical microscopic methods, and detection of specific cell wall components methods were used (*Joergensen & Wichern, 2008*). Also, the physiological method of selective respiratory inhibition, based on stimulation of respiration/metabolism of microorganisms by adding glucose to the substrate and subsequently inhibiting either fungi by adding cycloheximide or bacteria by adding streptomycin, was used (*Anderson & Domsch, 1975*). *Flinkman & Thörnquist (1986)* studied storage of bundled, unlimbed pulpwood and logging residues and found that the occurrence of microfungi in bundled material did not differ to any particular extent from loose material after eight months. They also noted that assortments with relatively large proportions of needles and bark appeared to provide the most favourable substrates for fungi. *Jirjis & Lehtikangas (1993)* studied fuel quality and dry matter loss during storage of logging residues in a pile. They found a general increase in spore count during approximately one year of storage and also an increase in viable spores. In another study by *Lehtikangas & Jiris (1995)* of logging residues in covered piles, the spore count at the end of the storage period (one year) was lower than at sampling after 7 months of storage, while the number of viable spores slightly increased. Spore counting does not discriminate between decay fungi and other fungal spores without additional isolation and identification in the lab. However, the development of DNA-based PCR (Polymerase Chain Reaction) and taxon-specific primers has provided a range of new possibilities. For example, *Piskur et al. (2011)* used PCR-DGGE method to analyze the fungal communities in degraded wood chips. Quantitative real-time PCR (qPCR) has proven to be a useful tool for the detection of plant pathogenic fungi and bacteria (*Hietala et al., 2009*; *Salm & Geider, 2004*; *Schaad & Frederick, 2002*; *Schena, Hughes & Cooke, 2006*; *Vandroemme et al., 2008*), estimation of fungal biomass in forest soil (*Baldrian et al., 2013*) but also wood deteriorating fungi (*Eikenes et al., 2005*; *Pilgård et al., 2011*; *Pilgård, Alfredsen & Hietala, 2010*; *Song et al., 2014*). The method is highly sensitive, specific and rapid, with the added capacity for quantification. To our knowledge, the qPCR approach has not so far been used to quantify the basidiomycetous fungi in logging residues.

The forestry practice of using the logging residues as a fuel chips is relatively recent and there is little available documentation on how microbial processes in stored piles may influence the final quality of the material as a fuel. Because the fungi degrade the wood and thus use up its energy, the amount of basidiomycetous DNA (indicative of the amount of fungal biomass) may be correlated with the degree of material degradation. Hence, we make the following assumption: the more basidiomycetous DNA measured, the higher the basidiomycetous colonization with degradation potential in the logging residue. By understanding the pattern of basidiomycetous colonization in stored logging residues

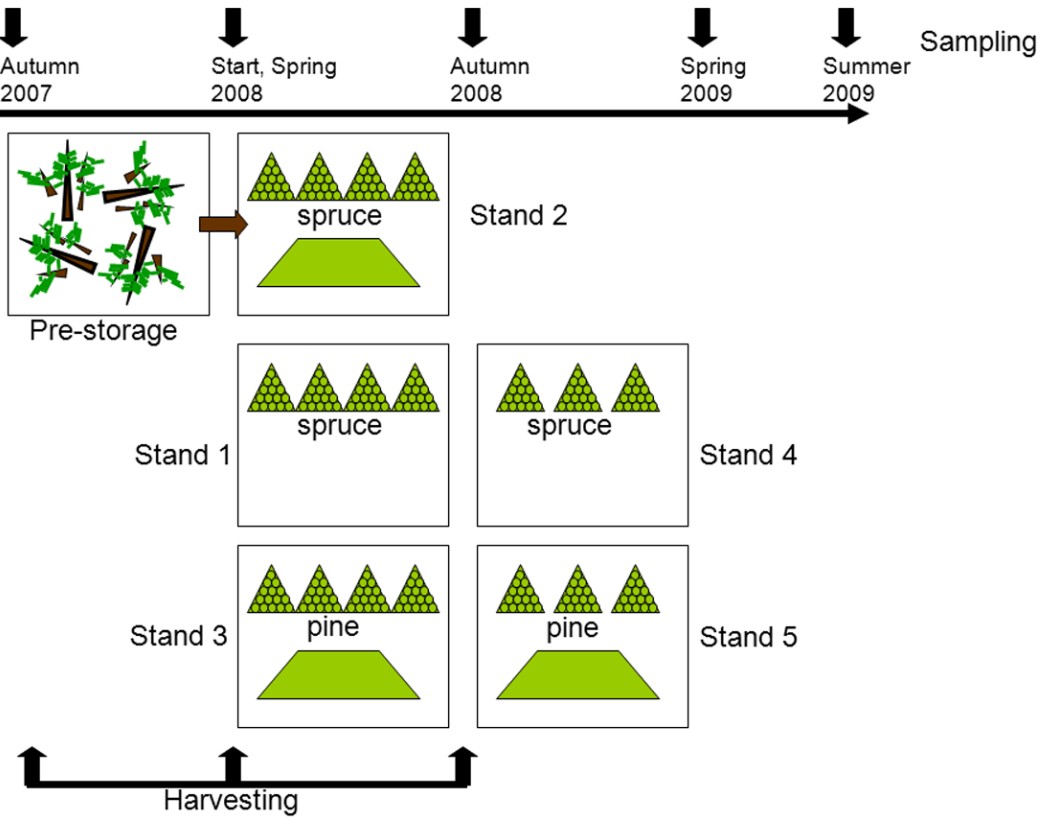

**Figure 1 Experimental setup.** After the first harvesting (felling of trees) the material was pre-stored by being left on the clear-cut during the winter. In the second and third harvesting periods piles of bundled (triangle symbol) and loose residues (trapezoid symbol) were constructed. At each sampling one bundled pile and a quarter of a loose pile were analyzed.

better, this knowledge can be used to (1) better understand the potential effect of storage on basidiomycetous colonization and (2) optimize storage methods further.

This paper is based on the same samples as described in *Filbakk et al. (2011)*. They modelled moisture content and dry matter loss during storage of logging residues. The first aim of this study was to implement quantitative qPCR as a novel technique to quantify the basidiomycete fungi in logging residues. The second aim was to find out how the storage conditions and type of forest residues influence the colonization by potentially decomposing basidiomycetous fungi.

## EXPERIMENTAL SECTION

### Experimental setup and sample taking

The detailed description of the experimental setup and sampling is given in *Filbakk et al. (2011)*. Briefly, this study was carried out with residues originating from five different harvesting sites, all located close to Braskereidfoss (60°62′N/12°02′E), Norway. Three stands were predominantly Norway spruce (*Picea abies* (L.) Karst.) and two were dominated by Scots pine (*Pinus sylvestris* L.), all 70–100 years old. The experimental setup is illustrated in Fig. 1. Stand 2 was harvested in the autumn 2007 and logging

residues were pre-stored; left lying on the clear-cut site until spring 2008 when piles were constructed. Stands 1 and 3 were harvested in spring 2008, and Stands 4 and 5 were harvested in autumn 2008. After each harvesting, the residues were stored in two types of piles, either in pyramid-like piles consisting of bundled residue material or loose piles consisting of unbundled material (Fig. 1). To protect the piles from precipitation, each pile was covered at the top with 2 mm thick residue wrapping paper produced by UPM Kymmene. At each sampling point the entire bundle was chipped in an industrial grinder (Peterson 4700B), resulting in about 1,000 kg of chipped material. From this source five replicates (1–2 kg each) were provided for further analysis. The chips were then removed before the next sample was chipped and sampled. Samples from piles were taken before storage (Start), then in spring 2008, autumn 2008, spring 2009, and summer 2009 (Fig. 1). All samples were analyzed for moisture content by the oven drying method at 103 °C (*CEN/TS-14774-1, 2004*), and for calorific value (*CEN/TS-14918, 2005*). To get a rough estimate of the dry matter loss in bundles, each bundle was weighed before placing into piles and at then before chipping by using a scale with accuracy of one kg. The total dry matter loss in bundled piles was calculated as the difference between initial and final mass of each bundle (*Filbakk et al., 2011*).

## DNA extraction

From each chipped sample, about 0.4 kg was randomly taken, dried at 50 °C for 24 h, and grinded first coarsely (Retsch Mühle grinder, 5 mm mesh; Retsch Gmbh, Haan, Germany) and then finely (IKA Werke MF 10 basic grinder, 0.5 mm mesh; IKA®-Werke Gmbh & Co., Staufen, Germany). The same sample batch was used for analyses by *Filbakk et al. (2011)*. About 80 mg of finely grinded material was milled to powder-consistency in liquid nitrogen using a Retsch mixer mill (MM 300, Retsch Gmbh, Haan, Germany). Aliquots of 20 mg were prepared from powdered material for each treatment and total DNA was extracted using a DNeasy Plant Mini Kit (Qiagen, Hilden, Germany). The protocol provided by the manufacturer was followed. To account for the variation in DNA extractability in the environmental samples and to normalize for this variation, 5 ng of an external reference DNA, pGEM plasmid (pGEM-3Z Vector; Promega, Madison, Wisconsin, USA), was added to each sample upon start of each DNA extraction (*Hietala et al., 2009*). The extracted DNA was eluted in 50 μl of buffer AE and stored at −80 °C until processed by qPCR.

## Primer selection

The primers were based on prior findings by *Vilgalys & Hester (1990)* and *Fierer et al. (2005)* with minor modification to improve the number of basidiomycetes amplified. As a forward primer the 5.8sr TCGATGAAGAACGCAGCG primer was used (*Fierer et al., 2005*; *Vilgalys & Hester, 1990*) and as a reverse primer we selected a 2 nucleotide truncated ITS4-X primer CAG GAG ACT TGT ACA CGG TCC as it amplifies a larger set of basidiomycetous species. Primers for the internal pGEM control for extractability were selected as previously described by *Coyne et al. (2005)* and *Pilgård, Alfredsen & Hietala (2010)*.

To test the primer specificity for basidiomycetes, pure cultures of organisms representing the target DNA (basidiomycetes) were used as positive controls (*Armillaria*

*borealis* 2005-713/2, *Fomitopsis pinicola* 1946-755/2, *Coniophora puteana* 1982-97/3 and *Schizophyllum commune* 1956-1236/1), and as negative controls non-target DNA (aseptic spruce seedling roots, *Trichoderma* sp. 1959-1919/471, *Penicillium* sp. 2007-161/14/3 and *Cladosporium cladosporoides* 1967-149/11), all fungi obtained from the Norwegian Forest and Landscape Institute's fungal culture collection (http://www.skogforsk.no/skogpatologi/database/searchform.cfm) were grown in petri dishes on cellophane over malt agar medium and incubated at 25 °C. To verify the specificity and sensitivity of the primers also on degraded wood samples, known dual mixtures of basidiomycetous DNA and host tree (*Serpula*-degraded and *Trametes*-degraded pine wood) together with internal control pGEM plasmid DNA were used. The presence of amplified qPCR product in the expected size range was verified by agarose gel runs. Negative controls consisted of both water and purified Scots pine DNA.

## qPCR conditions

The qPCR detection of basidiomycetous DNA ($DNA_{bas}$) was performed using SYBR Green PCR Mastermix (Applied Biosystems, Foster City, California, USA) and the reference pGEM was quantified with TaqMan Universal PCR Master Mix (Applied Biosystems #4304437; Applied Biosystems, Foster City, California, USA).

The internal standard pGEM for calibrating DNA extractability was quantified as described by *Coyne et al. (2005)*. The pGEM standard curve was prepared from serial diluted samples containing 0.066–0.000066 ng of pGEM DNA giving a standard curve of $y = 2.7371 − 0.2642x$ ($x$ is the Cq value and $y$ the logarithmic amount of pGEM DNA present in the sample).

For quantification of $DNA_{bas}$ a standard curve was prepared from DNA isolated from four basidiomycetous fungi (*A. borealis*, *F. pinicola*, *C. puteana* and *S. commune*), using serial diluted samples of 0.03–0.0003 ng of DNA giving a standard curve of $y = 3.0922 − 0.176x$ ($x$ is the Cq value obtained by qPCR and $y$ the logarithmic amount of $DNA_{bas}$ present in the sample).

For qPCR detection and quantification of $DNA_{bas}$ the primer concentrations of 60 nM and 3 μl of extracted DNA solution from each sample were used for each reaction. The qPCR cycling was carried out using cycle parameters of 95 °C for 10 min to activate the polymerase, followed by 40 amplification cycles of 95 °C for 15 s and 60 °C for 1 min with signal thresholds set automatically.

The qPCR was performed with ABI PRISM 7700 (Applied Biosystems Foster City, California, USA). After amplification the data were analyzed and plotted (fluorescence vs. cycle number) using the Sequence Detection System, version 1.7a, Software Package (Applied Biosystems Foster City, California, USA). The extent of amplification was calculated as a mean Cq value of 2 technical replicates for each sample. All PCR reactions were performed in singleplex conditions under standard PCR cycling parameters. Undiluted, $10\times$, $100\times$, $1,000\times$ and $10,000\times$ diluted experimental sample concentrations were tested to investigate the presence of compounds inhibitory (impurities) to PCR amplification and to ensure that the amount of template fell in the linear range of the

**Table 1** Description of parameters used in the study.

| Parameter | Type | Explanation | Unit |
|---|---|---|---|
| Harvesting time | categorical | Season when the trees were cut (harvested) | Start, spring, autumn |
| Storage time | categorical | Days after the piles were made | Days |
| Precipitation | continuous | Mean precipitation since last sampling | Mm |
| Temperature | continuous | Mean temperature within piles | °C |
| Wood moisture | continuous | Moisture content measured in the sampled material | % |
| Tree species | categorical | Dominant tree species in the logging residue | Spruce, pine |
| Storage method | categorical | Piles made of loose or bundled forest residues | Start, loose, bundles |
| Calorific value | continuous | The amount of energy per kg given off when burnt | MJ/kg |
| Total dry matter loss | continuous | Total loss of dry matter during the storage, including fall off needles, twigs and microbial degradation | % |
| Placement | categorical | Location of bundles within piles | Top, bottom, middle |
| Pre-storage | categorical | Leaving the logging residue spread on the clear-cut site after harvesting, throughout the winter | Pre-storage, piles |

standard curve. The $100\times$ dilution showed no indications of inhibition and will thus be presented. Five biological and two technical replicates were used for each sample.

## Statistical analysis

We analyzed the amounts of $DNA_{bas}$ in logging residues from five forest stands as dependent parameter against the following independent categorical parameters: tree species, harvesting times, storage, within bundle placement, storage time. Precipitation, moisture, bundling time, mean temperature, dry matter loss and calorific value were continuous parameters in the model (Tables 1 and 2). The values for $DNA_{bas}$ were normally distributed. To test whether the parameters had significant effects on the amount of $DNA_{bas}$, we used analysis of variance (ANOVA) for all categorical parameters and linear regression for all continuous parameters. In addition, we tested interaction among wood moisture and storage method. First we tested whether pre-storage (leaving the logging residues on the harvesting site during the winter) had a significant effect on the $DNA_{bas}$ between Stands 1 and 2. Because it did not, we included all five stands in the same model. Means in categorical parameters were compared by Tukey–Kramer HSD test at $P < 0.05$). In all cases, a null hypothesis was rejected at the 5% level of significance. All statistics were done using JMP (SAS Institute Inc., Cary, NC, USA).

## RESULTS AND DISCUSSION

### qPCR assay

The qPCR primers used in our assay were specific for tested $DNA_{bas}$, while the non-target DNA used as negative controls (water, host tree DNA, bacterial and non-basidiomycetous fungi) was not amplified under the conditions used. Likewise, the primers used for the calibration of DNA extractability between samples amplified only the pGEM plasmid used as an internal control in these studies. The qPCR can detect down to one average size fungal

**Table 2  Test statistics and parameter estimates of the model.**

|  | Variance components: |  |
|---|---|---|
| $R^2$ | 0.68 |  |
| $R^2$ adj | 0.65 |  |
| RMSE (Root Mean Square error) | 0.10 |  |
| N | 133 |  |
| **Parameter estimates for the covariates in the model and *P*-values from effect tests:** | | |
|  | **Parameter estimate** | ***P*-values** |
| Intercept | 0.3350 | 0.0024[*] |
| Storage time | −0.0004 | <.0001[*] |
| Precipitation mm | 0.0333 | <.0001[*] |
| Storage method | – | <.0001[*] |
| Mean temperature | −0.0054 | 0.0040[*] |
| Harvesting time | – | 0.0061[*] |
| Tree species | – | 0.0091[*] |
| Wood moisture*storage method | – | 0.0397[*] |
| Wood moisture | −0.0045 | 0.0398[*] |
| **Parameter estimates for categorical variables and Tukey–Kramer (T–K) 0.05 tests on LS means:** | | |
|  | **Parameter estimate** | **T–K** |
| Harvesting time [Autumn 2007] | 0.0301 | A |
| Harvesting time [Spring 2008] | 0.0218 | A |
| Harvesting time [Autumn 2008] | −0.0519 | B |
| Tree species [pine] | 0.0351 | A |
| Tree species [spruce] | −0.0351 | B |
| Storage method [Start] | −0.1393 | C |
| Storage method [Bundle] | 0.0334 | B |
| Storage method [Loose] | 0.1059 | A |
| Wood moisture*storage method [Start] | 0.0079 | – |
| Wood moisture*storage method [Bundle] | −0.0026 | – |
| Wood moisture*storage method [Loose] | −0.0053 | – |

**Notes.**

[*] Significance level 0.05.

genome (i.e., one fungal cell) corresponding to the lowest point on our standard curve (∼0.0003 ng).

Our qPCR assay was able to quantify $DNA_{bas}$ in logging residues. It detected $DNA_{bas}$ in the range of 0.01–0.69 ng $DNA_{bas}$/mg in chipped logging residue (Fig. 2). That the most elevated amounts of $DNA_{bas}$ detected was after the highest precipitation event i.e., conditions promoting basidiomycetous growth (see below, 'Variation in $DNA_{bas}$ over time' and 'Moisture') is an indicator that the sampling was representative and that the selected primers were efficient. This same general pattern was found in all stands: the lowest $DNA_{bas}$ value was detected at the start of the storage period, followed by increase in the wetter period in spring 2009, followed by a decrease during the dryer period at the end of the storage, suggesting that our detected values may reflect the environmental conditions impact on basidiomycetous growth. Our $DNA_{bas}$ values were considerably higher than those found by *Pilgård et al. (2011)* in Scots pine heartwood, preservative treated and

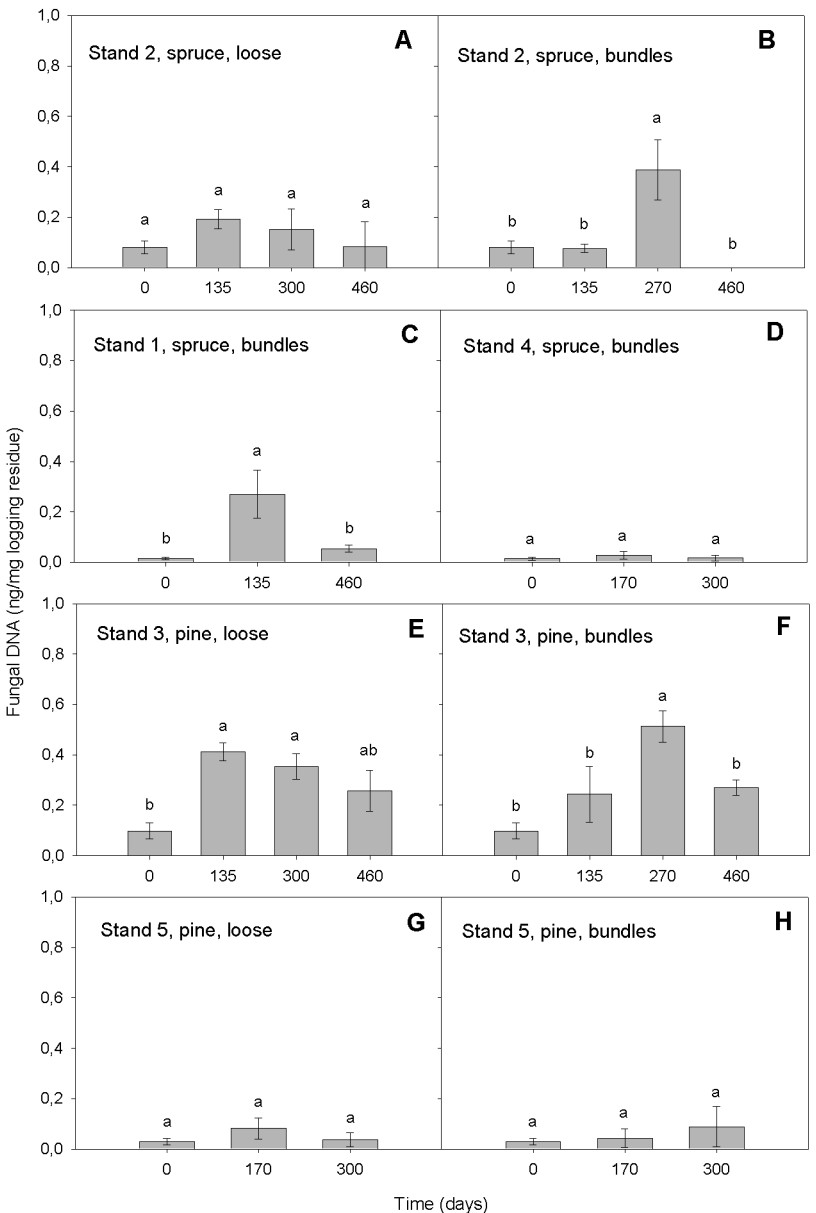

**Figure 2 Amount of DNA$_{bas}$ (ng/mg logging residue) detected in logging residue after increasing storage time (days) in all five stands, in loose and bundled piles of spruce and pine ($n = 133$).** Standard errors are marked as vertical lines and means are compared by Tukey–Kramer test at $P < 0.05$, where different letters denote significant differences.

furfurylated stakes in a soil contact (EN 252, 1989), when using qPCR. However, the substrates and exposure situation in these two studies were very different; Scots pine heartwood has some natural resistance to fungal decay and preservative treated or modified wood are commercial treatments used to hinder the basidiomycetous deterioration. In this study, the forest residues were not treated with wood protection systems in order to prolong their service life and thus were likely exposed to higher colonization potential by fungi, which may explain higher DNA$_{bas}$ in samples tested in our study.

Although the qPCR assays are highly sensitive with high resolution and correlation to mass loss at the early stages of decay, they may lack accuracy in advanced decay stages due to substrate depletion (*Eikenes et al., 2005*). We did not have substrate depletion in our material, since the level of decay was at all times only in its initial stages of colonization and thus substrate availability was plentiful for further basidiomycetous growth when the environmental conditions were favorable.

The accuracy of the qPCR based quantification of fungal biomass depends on extraction efficiency and purity of extracted DNA. The mean and modus concentrations of extracted DNA in our samples were even 95 and 83 ng/μl, respectively. The ratio of absorbance at 260 nm and 280 nm is used to assess the purity of DNA. The mean and modus value for 260/280 ratio in our extracted DNA measured by NanoDrop (NanoDrop Technologies, Inc, Wilmington, Delaware, USA) was 1.8, which is accepted as "good quality" DNA. The absorbance ratio 260/230 is used as a secondary measure of nucleic acid purity and expected values are commonly in range 2.0–2.2. Mean and modus values in our samples were 1.4, suggesting that contaminants absorbing at 230 nm, such as carbohydrates or phenols, may have been present. Impact of contaminants was avoided by diluting samples, and dilution 100X ensured optimal accuracy.

In spite of the increasing use of qPCR for fungal biomass estimation, it is known that there is variation in ITS copy numbers (per ng DNA) among fungal species (*Baldrian et al., 2013*; *Song et al., 2014*), which will impact on the biomass estimation. We used a DNA from four different basidiomycetes to make our standard curves to partly accommodate for such ITS differences, but the obtained values must be considered as estimates of $DNA_{bas}$. When *Baldrian et al. (2013)* compared fungal biomass content in litter and soil by using three different methods (ergosterol, phospholipid fatty acid and qPCR), they demonstrated that the obtained estimates led to large differences in relative amount of litter and soil fungal biomass. They estimated 28, 7 and 2 times larger fungal biomass in litter than soil by ergosterol, phospholipid fatty acid and qPCR, respectively, indicating that qPCR was the one least likely to overestimate the fungal biomass in litter.

### $DNA_{bas}$ and storage conditions

The statistical model we used included data from all stands and explained 68% of the variability in the data (Table 2, $n = 133$, $R^2 = 0, 68$). Of all the parameters we tested, only harvesting time, storage time, precipitation, temperature, wood moisture content, tree species and storage method significantly affected the amount of $DNA_{bas}$ detected in the logging residues (Table 2). In addition, the wood moisture*storage method contributed significantly. As fungi colonize the material, fungal biomass, species diversity and material decomposition change. Therefore in our study, each time the sample was taken, offers a mere snapshot of the situation at the given time.

### *Variation in* $DNA_{bas}$ *over time*

In the model, the storage time affected the $DNA_{bas}$ amount significantly ($p < 0.0001$). Generally, the amount of $DNA_{bas}$ decreased at the end of the storage (Figs. 2 and 3). In all stands and pile types we detected the lowest $DNA_{bas}$ values at the start, slightly rising

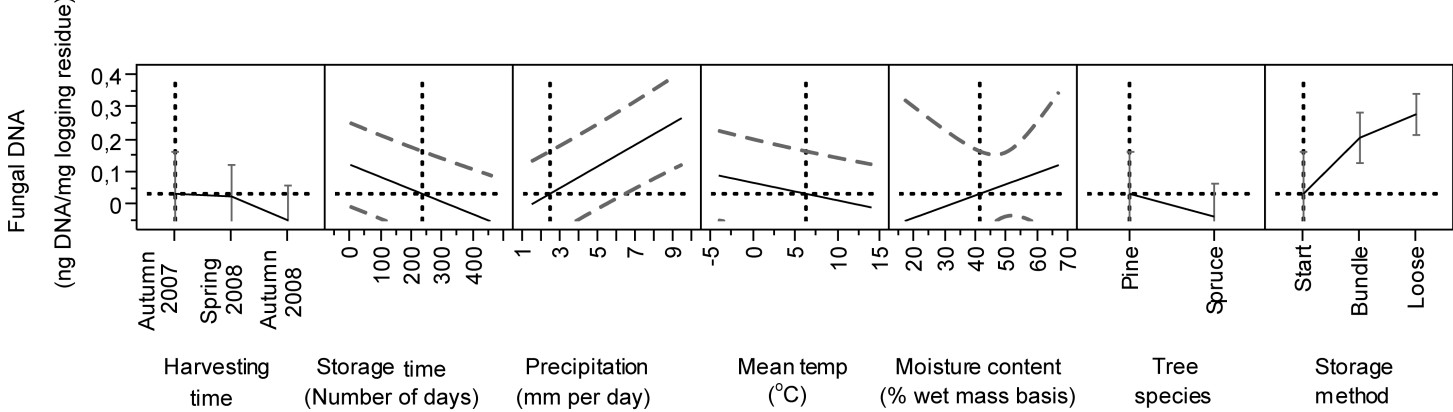

**Figure 3 Predicted estimates of DNA$_{bas}$ in logging residues (ng DNA/mg logging residue) in relation to harvesting time, storage time, precipitation, temperature, moisture content, tree species and storage method.** Vertical and horizontal dotted lines show the current values for variables on *x* and *y*-axes. The black lines show how the predicted value of the variables on the *y*-axes change with changing of the values on the *x*-axes. The 95% confidence interval for the predicted values is shown by striped line surrounding the prediction trace (or error bar for categorical variables harvesting time, tree species and storage method).

during the storage and decreasing at the end of the study (Fig. 2). The transient DNA$_{bas}$ increase was significant for the combinations illustrated in Figs. 2B, 2C, 2E and 2F.

The highest DNA$_{bas}$ content was detected in bundles after 270 days, in spring 2009, in a Norway spruce stand (*n* = 5, mean 0.39 ng DNA, Fig. 2B) and in a Scots pine stand (*n* = 5, mean 0.51 ng DNA, Fig. 2F). Both peaks coincided with the highest precipitation (9 mm) measured during the entire study period, almost 5-fold higher than the mean precipitation.

The general decline in amount of DNA$_{bas}$ with time may be due to transpiration drying, as transpiration continues from foliage or open wood surfaces after harvesting (*Andersson et al., 2002*). Indeed, such a decline in moisture has already previously been documented in this material by *Filbakk et al. (2011)*.

Material from Stands 4 and 5, with the lowest DNA$_{bas}$ values, was harvested and piled in autumn 2008 and had the shortest storage period. It is likely that the conditions were not conducive for fungal development during most of the storage period (winter/spring), since basidiomycetous fungi have restricted growth at low temperatures.

### Seasonal effects

The model showed a significant contribution of the harvesting time (time of felling), to the amount of DNA$_{bas}$ (Table 2). We detected significantly lower start DNA$_{bas}$ values from material harvested in autumn 2008 than in autumn 2007 and spring 2008 (Table 2 and Fig. 3). The moisture content in the stored biomass tends to increase in late autumn and winter in the Nordic climate (*Nurmi, 1999*; *Pettersson & Nordfjell, 2007*). Although studies by *Nurmi & Hillebrand (2007)* showed that harvesting in spring gave better conditions for drying, thus less favorable conditions for fungi, and is in line with our observations.

### Moisture

In the model, wood moisture ($p = 0.0398$), precipitation ($p < 0.001$) and the wood moisture*storage method ($p = 0.0397$) contributed significantly in explaining $DNA_{bas}$ content (Table 2). The prediction profiles illustrate that $DNA_{bas}$ increased with precipitation and wood moisture content. We found the highest values of $DNA_{bas}$ in material with moisture 30–35% (Fig. 3). In general, as expected, the moisture content decreased during storage for all materials, in Stand 2 from 55 to 50%, Stands 1 and 3 from 45% to 30%, Stand 4 from 45 to 40% and Stand 5 from 55 to 40% (*Filbakk et al., 2011*). The average moisture content reduction was 12% larger in summer than in winter, the relative air humidity in the piles was about 100 % and the temperature in the piles closely followed the ambient temperature, due to fairly small piles, never exceeding 19 °C (*Filbakk et al., 2011*). The significant interaction of the moisture*storage method may be because the "Start" was included in our study as one of the storage methods (Table 2 and Fig. 3) and the moisture content was highest at this point.

Our results indicate that the moisture content in the piles was sufficient for growth of basidiomycetous fungi, never below 20%. A wood moisture content of 20% is often used as a threshold value for risk of fungal growth (*Zabel & Morrell, 1992*). Water is needed for fungal decay of wood because it (1) participates at hydrolysis, (2) serves as a diffusion medium, (3) is a solvent or life-supporting medium, (4) a mean for capillary swelling of wood (*Zabel & Morrell, 1992*). However, fungi can survive in a dormant state and rewetting will revive them (*Findlay, 1950*). In general, optimal moisture for growth of the majority of wood decaying basidiomycetes is in the range between 40 and 80% (*Eaton & Hale, 1993*).

Indeed, the high amount of precipitation (9 mm) in spring 2009, after 270 days of storage, was associated with higher $DNA_{bas}$ values (Figs. 2B and 2F). The temporary peak $DNA_{bas}$ values related to peak moisture in the material (*Filbakk et al., 2011*) further indicate that even a short period of high precipitation may increase the basidiomycetous growth.

### Tree species

Tree species contributed significantly ($p = 0.0091$) to the amount of $DNA_{bas}$ in logging residues (Table 2). The Tukey–Kramer test showed a significant difference between Scots pine and Norway spruce (Table 2). We found significantly more $DNA_{bas}$ in pine-dominated residues ($n = 64$, mean 0.19 ng $DNA_{bas}$) than in spruce-dominated ones ($n = 69$, mean 0.10 ng $DNA_{bas}$), and there was larger variation in values in pine than in spruce residues (Figs. 2 and 3).

The significant differences between the tree species were mainly due to the high $DNA_{bas}$ values we detected in pine stand 3. Pine stand 5 had very low values. Because the spruce has a higher percentage of small branches with needles the bundles are usually denser, with more moisture and less airflow, than pine bundles. The environment in pine bundles may have been more conducive to basidiomycetous growth than in spruce bundles.

The high variability among the $DNA_{bas}$ found at the five different stands, with high values in pine stand 3 (mean 0.5 ng $DNA_{bas}$) and correspondingly low values at spruce

stand 4 (mean 0.02 ng $DNA_{bas}$), probably contributed to the significant differences among the tree species.

### Storage method

The storage methods alone ($p < 0.0001$) and also in combination with moisture ($p = 0.0397$) contributed significantly in explaining the amount of $DNA_{bas}$ in logging residues (Table 2). We found significant differences in $DNA_{bas}$ among the three storage methods. In all stands the amount of $DNA_{bas}$ was, as expected, always lowest at start, before bundling or loose pile construction ($n = 20$, mean 0.04 ng $DNA_{bas}$). Compared to start, the $DNA_{bas}$ was significantly higher in bundles ($n = 65$, mean 0.16 ng $DNA_{bas}$) and highest in loose residues ($n = 48$, mean 18 ng $DNA_{bas}$).

Only minor differences were detected in the moisture content between bundles and loose piles (Filbakk et al., 2011). Hence the consistence of loose piles may have created microclimate with more air flow, thus more conducive to basidiomycetous development than in the compacted bundles.

We did not find any significant differences in $DNA_{bas}$ amount related to material location within the pile. Filbakk et al. (2011) found the highest material moisture in the middle of the pile; however, the moisture content in this study was always within the favorable range for basidiomycetous growth (Eaton & Hale, 1993).

When comparing bundled spruce material, with or without pre-storing in Stands 1 and 2, we found that the mean $DNA_{bas}$ content was similar, 0.11 and 0.13 ng $DNA_{bas}$ respectively ($n = 42$, Figs. 2B and 2C). This may imply that the pre-storage on the harvesting site during the winter, before piling, does not increase the decay of the logging residues.

### Practical implications

Both drying (including fall off needles and twigs) and microbial activity will result in dry matter loss in logging residues. In Filbakk et al. (2011) about 20% total dry matter loss was measured for bundled residues. However, this dry matter loss was most probably caused by handling together with the process of transpiration drying, causing the dry foliage and small twigs to fall off. Because the error in exact measuring of the dry matter loss in bundles was large, the mass loss caused by fungi alone could not be specified. The needle and twig fall off is considered beneficial because (1) its presence lowers the final combustion efficiency and (2) the nutrients contained in this material remain at the site as opposed to nutrient depletion by its complete removal.

We show that moisture is a key factor for basidiomycetous growth in piles. During storage time of maximum 460 days the moisture content in logging residues gradually decreased together with $DNA_{bas}$ content. Hence, our results indicate that that storage of logging residues for maximum 460 days under these conditions give low levels of decay. Because we found higher values of $DNA_{bas}$ in loose residues compared to bundles, the practice of bundling does not seem to promote more basidiomycetous growth than leaving the residues in loose piles. Neither did pre-storing of residues during winter seem to make a difference to $DNA_{bas}$ content.

Our data suggest that basidiomycetous growth may be temporarily stimulated when the material has the optimal water content, such as in periods of increased precipitation, This supports the practice of covering the piles during storage.

## CONCLUSIONS

- We show that qPCR assay to quantify the $DNA_{bas}$ is a method, capable of detecting the basidiomycetous biomass in logging residue material.
- Because the $DNA_{bas}$, indicating the presence of potentially decomposing basidiomycetes, decreases at the end of the storage period together with moisture content, there seems to be a small danger of fungi significantly transforming logging residues stored for less than 460 days at given conditions.
- Scots pine residues had more $DNA_{bas}$ than Norway spruce residues.
- Loose piles had generally more $DNA_{bas}$ than bundled ones.

## ACKNOWLEDGEMENTS

We wish to thank Inger Heldal, Kari Hollung, Sigrun Kolstad, and Eva Grodås for their excellent technical help and Nina E. Nagy for her help in using the SigmaPlot programme.

### Funding

We are grateful to the Research Council of Norway for their financial support in project no. 315052. Material sampling was financed through CenBio. The funders had no role in study design, data collection and analysis, decision to publish, or preparation of the manuscript.

### Grant Disclosures

The following grant information was disclosed by the authors:
Research Council of Norway: 315052.
CenBio.

### Competing Interests

All the authors are employees of the Norwegian Forest and Landscape Institute.

### Author Contributions

- Isabella Børja and Gry Alfredsen conceived and designed the experiments, performed the experiments, analyzed the data, contributed reagents/materials/analysis tools, wrote the paper, prepared figures and/or tables, reviewed drafts of the paper.
- Tore Filbakk conceived and designed the experiments, performed the experiments, analyzed the data, contributed reagents/materials/analysis tools, prepared figures and/or tables, reviewed drafts of the paper.
- Carl Gunnar Fossdal conceived and designed the experiments, performed the experiments, analyzed the data, contributed reagents/materials/analysis tools, reviewed drafts of the paper.

## Supplemental Information

Supplemental information for this article can be found online at http://dx.doi.org/
10.7717/peerj.887#supplemental-information.

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
