# Peer review of "DNA quantification of basidiomycetous fungi during storage of logging residues"

_PeerJ, doi:10.7717/peerj.887_

## Round 0.1 · original submission · Major Revisions

· Academic Editor

Major Revisions

The reviewers (especially reviewer #2) pose some significant questions regarding some of the quantitative elements of the study and make some fairly detailed recommendations on how the manuscript could be improved. Please try to address these in your revision.

·

Basic reporting

I feel that this paper is an acceptable publication and does contribute to the body of literature on degradation of logging residues. I have some apprehension about the assumption that fungal DNA=more dry matter loss. If you are quantifying DNA in logging residues, you are at best getting a snapshot of what is there, but has doesnt give you the whole story on whats actually degrading the wood. That requires a whole other set of tools (gene expression, metabolite screening, wood decomp. measures, etc.) That aside, i think that this is a good paper and should be acceptable for publication in this outlet.

Experimental design

Well presented and detailed. The author has done a nice job of describing what was done.

Validity of the findings

Good. I wonder how informative it would be to pair these results with bacterial data to determine what the rate limiting factors are in this system. I think fungi are probably contributing greatly to the breakdown of the wood biomass, but bacteria have been shown to increase porosity of wood cell wall through breakdown of pit membranes, etc. and could be effecting drying rates that could in turn affect fungal colonization.
Also, two additional things were not clear to me after reading this paper, (1) the role of soil arthropods in the process and (2) if the piling process presents enough biomass to promote thermophilic fungi from composting the material.You might want to at least mention how bugs and composting might impact this system.

Reviewer 2 ·

Basic reporting

According to PeerJ guidelines, the submission must adhere to all PeerJ policies. Since I do not dispose with all information connected to this submission, I cannot fully confirm that the submission fulfills all PeerJ requirements (eg. Author policies, Publishing plan policies). Nevertheless, I believe editor(s) have the information needed and can decide if the submission adheres to all PeerJ policies.

As a reviewer, I would alert on the following issues:

- language corrections (eg. line125-126 -> rephrase “total DNA extracted”; line 162 -> rephrase “sample extracted total DNA solution”; line 194-195 -> rephraase the sentence, …)
- please unify the phrases throughout the text (eg. use for example “basidiomycetous DNA” when reffering to the DNA that was measured in your study)
- also, please be accurate with your statements (eg. L. 188: “… were specific for TESTED basidiomycetous DNA…”; L. 196-198: “that the method was sufficiently sensitive FOR WHAT”; L.207-208: “…which can explain higher BASIDIOMYCETOUS DNA IN SAMPLES, TESTED IN OUR STUDY”; L.214: please specify what DNA (eg. basidiomycetous DNA detected with qPCR…); please check the text for other examples.
- one of the PeerJ policies is “Discipline Specific Standards”, where following is stated “Reports utilizing quantitative real time PCR should follow the MIQE guidelines (the Minimum Information for Publication of Quantitative Real-Time PCR Experiments) and checklist”. I would suggest that authors check these guidelines and cite and supplement the manuscript with the missing essential information (eg. PCR conditions, calibration curves)
- authors have made their main assumption for the manuscript on the citation Eikenes et al., 2005 (lines: 77-80) (“the more fungal DNA measured, the more dry matter loss”). Nevertheless, Eikenes et al. findings are not in concordance with the authors assumptions, quite on contrary (Eikenes et al, 2005 state: DNA-quantification is not suited for estimating mass loss after 16-20 weeks after inoculation). I am aware that the exp conditions differ between these two experiments and also that authors later (lines:209-213) mention the correlation in advanced decay stages, but I would strongly recommend to include other studies and / additional info to back-up the authors’ assumption.
- I would also suggest to include some latest articles about qPCR for measuring biomass of fungi in plant/wood material (eg. Song et al. 2014: Quantitative PCR for measuring biomass of decomposer fungi in planta; Baldrian et al. 2013: Estimation of fungal biomass in forest litter and soil) and to stress out the main concerns of using qPCR for estimation of fungal biomass and its activities…

Experimental design

Following suggestions should be addressed:
- as I understand the research described in the manuscript was done in the same experimental setup & design as in Filbakk et al. (2011). Nevertheless, I would strongly advise that authors give more detailed descriptions of the experiments and avoid refering to Filbakk et al. (2011) in such an extent as it is in the current form of the manuscript. I, as a reader, would be grateful if the sampling methodology would be more precisely described (eg. when was the material chipped, how was the dry matter loss calculated – based on subsamples of the piles or weighing the whole pile; what is the error of the dry matter estimation – the biomass is given per dry mass of what (dry matter, including needles and wood and other inpurities from the piles?) – can we compare these data, knowing that there is a substantial mass loss during storage? (comparing to eg. sampling techniques based on volume – see also Song et al. 2014). Also, how was the concentration of basidio DNA calculated – on what lodgging residues (dry mass / with or without needles / …)
- number of replicates per DNA extraction – what is the influence of the extraction step on the final results; what was the total DNA concentration in the extracted samples (not just the “basidiomycetous”)
- please provide the results for calibrating DNA extractability with the pGEM
- provide calibration curves (maybe as a Suplement material)
- give more detail about the PCR conditions
- give the number of replicates for analyses of fungal DNA. I would suggest to give the information about the replicates more clearly somewhere in the “Experimental Section”…
- statistical analyses: please, give more data how you performed the analyses – have you merged all data (storage type, tree species, ) for testing eg. harvesting time (Fig. 3)

Validity of the findings

- line 175: what does “approximately” means?
- line 192-193: please explain or insert a citation for the last sentence
- differences in DNA in different tree species – what about the inhibitors present in different tree species, extraction protocol, sample handling… ? Can you discuss also these factors? pGEM results would help to support your findings here.
- lines 209-213: I would suggest to expand this discussion with other studies, also the drawbacks of qPCR should be stressed out and discussed together with the obtained results. Also, the authors state, that the material was in its initial stages of colonization – please give support for this sentence.
- the manuscript main assumption is “more fungal DNA measured, the more dry matter loss”. But, the results of dry matter loss are not clearly demonstrated & presented.
- standard curve was prepared from DNA of 4 fungal species. As I understand, authors correlated the fungal (basidio) DNA in the samples to this standard curve. It is known, that fungal species and isolates differ in their genome size and ITS copy numbers and DNA quantification can underestimate fungal biomass (se Baldrian 2013). Most likely fungal communities in the piles under study differ in structure and developed differently. It is so questionable, if the measuring of DNA in such diverse material can be compared as it is in this manuscript. I would suggest, that authors dicuss also this aspects of the used methodology.
- as also Filbakk et al (2011) pointed out – the measurement method for determination of dry mass is approximate (mass loss during bundle handling, …) and also increase in dry mass was reported. Also, differences in mass losses between tree species are atributed to fallen needles – according to all stated – is comparison between basidiomycetous DNA measured in two tree species really possible? (if we know that the unit of dry mass differs (eg. more needle residues; errors with calculating dry mass…)?
- L. 323-329: it is difficult to talk about dry matter loss as a consequence of fungal activities, when the error measuring the dry matter loss is high. It is evident, that physical losses are the main contributors to mass loss (needles, branches, losses of material at handling processes of bundles) – couldn’t these “overwrite” the mass losses induced by fungi? Are there any known data about the mass loss of wood (spruce, pine) by fungal degradation in nature? Of course, every natural environment would give different estimates for this, but would be better to compare to, than laboratory estimates.
- Line 333-334: Please, rephrase the sentence. It is more than speculative to state, that your study “indirectly” showed that fungal DNA couldn’t cause this loss alone.
- line 335: what does “substantial” mean? Please, be more precise.
- Line 336-337: it is evident, also from the previous publication that the main part of dry matter loss is due to physicall losses, this study and the results from this study don’t show that. What is shown is, that losses are the sum of physical losses and biological degradation. If you would have a ratio between these two, you could discuss in more detail biological losses in your article.
- L. 343-344: Please, be precise – does your results really show, that there is no influence of fungal activities on fuel quality?
- L. 352-353: I strongly advise not to state such sentences, since they can have an impact on forest management policies – and your data are not strong supporters for statements like these…
- Line 87-89: one of the aims was to “find out whether the storage conditions of forest residues promote fungal growth to such a degree that it will significantly impair the final dry mass and fuel quality of the logging residues”. Nevertheless, there are no results and no discussion on this topic – especially about fuel quality.

My main concerns about this article are the errors in calculting / measuring the dry matter losses. But, nevertheless, data are still data and as I believe the policy of PeerJ is, that negative / inconclusive results are acceptable – I suggest that authors rewrite the article in a way to be more precise in their conclusions and that speculations are clearly presented as speculations (!). I would strongly suggest to present the errors and data for dry matter losses, fuel quality, results from qPCR, …

Additional comments

My main concerns about this article are the errors in calculting / measuring the dry matter losses. But, nevertheless, data are still data and as I believe the policy of PeerJ is, that negative / inconclusive results are acceptable – I suggest that authors rewrite the article in a way to be more precise in their conclusions and that speculations are clearly presented as speculations (!). I would strongly suggest to present the errors and data for dry matter losses, fuel quality, results from qPCR, …

---

## Round 0.2 · accepted · Accept

· Academic Editor

Accept

I believe you have addressed the concerns of the reviewers and improved the presentation of your research.